# Diagnostic and antibiotic use practices among COVID-19 and non-COVID-19 patients in the Indonesian National Referral Hospital

Robert Sinto[1,2,3,4]*, Khie Chen Lie[1,2], Siti Setiati[4,5], Suhendro Suwarto[1], Erni J. Nelwan[1,2], Mulya Rahma Karyanti[2,6], Anis Karuniawati[2,7], Dean Handimulya Djumaryo[2,8], Ari Prayitno[2,6], Sumariyono Sumariyono[4,9], Mike Sharland[10], Catrin E. Moore[10], Raph L. Hamers[3,11], Nicholas P. J. Day[3,12], Direk Limmathurotsakul[3,12,13]

1 Division of Tropical and Infectious Diseases, Department of Internal Medicine, Cipto Mangunkusumo National Hospital, Faculty of Medicine Universitas Indonesia, Jakarta Pusat, Jakarta, Indonesia, 2 Infection and Antimicrobial Resistance Control Committee, Cipto Mangunkusumo National Hospital, Jakarta Pusat, Jakarta, Indonesia, 3 Centre for Tropical Medicine and Global Health, Nuffield Department of Medicine, University of Oxford, Oxford, United Kingdom, 4 Department of Internal Medicine, Cipto Mangunkusumo National Hospital, Faculty of Medicine Universitas Indonesia, Jakarta Pusat, Jakarta, Indonesia, 5 Center for Clinical Epidemiology and Evidence Based Medicine, Faculty of Medicine Universitas Indonesia, Cipto Mangunkusumo National Hospital, Jakarta Pusat, Jakarta, Indonesia, 6 Department of Child Health, Cipto Mangunkusumo National Hospital, Faculty of Medicine Universitas Indonesia, Jakarta Pusat, Jakarta, Indonesia, 7 Department of Clinical Microbiology, Faculty of Medicine Universitas Indonesia, Cipto Mangunkusumo National Hospital, Jakarta Pusat, Jakarta, Indonesia, 8 Department of Clinical Pathology, Cipto Mangunkusumo National Hospital, Faculty of Medicine Universitas Indonesia, Jakarta Pusat, Jakarta, Indonesia, 9 Board of Directors, Cipto Mangunkusumo National Hospital, Jakarta Pusat, Jakarta, Indonesia, 10 Centre for Neonatal and Paediatric Infection, St George's University of London, Cranmer Terrace, London, United Kingdom, 11 Oxford University Clinical Research Unit Indonesia, Faculty of Medicine Universitas Indonesia, Jakarta Pusat, Jakarta, Indonesia, 12 Mahidol Oxford Tropical Medicine Research Unit, Faculty of Tropical Medicine, Mahidol University, Bangkok, Thailand, 13 Department of Tropical Hygiene, Faculty of Tropical Medicine, Mahidol University, Bangkok, Thailand

* robert.sinto01@ui.ac.id, robert.sinto@ndm.ox.ac.uk

**Data Availability Statement:** All relevant data are available from within the paper and Supporting information files. We permit the use of third party data not contained within the paper, available from

## Abstract

### Background

Little is known about diagnostic and antibiotic use practices in low and middle-income countries (LMICs) before and during COVID-19 pandemic. This information is crucial for monitoring and evaluation of diagnostic and antimicrobial stewardships in healthcare facilities.

### Methods

We linked and analyzed routine databases of hospital admission, microbiology laboratory and drug dispensing of Indonesian National Referral Hospital from 2019 to 2020. Patients were classified as COVID-19 cases if their SARS-CoV-2 RT-PCR result were positive. Blood culture (BC) practices and time to discontinuation of parenteral antibiotics among inpatients who received a parenteral antibiotic for at least four consecutive days were used to assess diagnostic and antibiotic use practices, respectively. Fine and Grey subdistribution hazard model was used.

Hospital Information System Management Cipto Mangunkusumo National Referral Hospital: hospital admission, microbiology, and drug dispensing data set that could be accessed through umsi@rscm.co.id.

**Funding:** This study is funded by Research Fund of Division of Tropical and Infectious Diseases, Department of Internal Medicine, Cipto Mangunkusumo National Hospital and in part by the Wellcome Trust [220211]. RS is funded by Indonesian Education Scholarship (Beasiswa Pendidikan Indonesia [BPI]) [202101182688] from the Ministry of Education, Culture, Research, and Technology Republic of Indonesia: Directorate General of Higher Education, Research, and Technology (Direktorat Jenderal Pendidikan Tinggi, Riset dan Teknologi Kementerian Pendidikan, Kebudayaan, Riset dan Teknologi Republik Indonesia) and Indonesian Endowment Fund for Education (Lembaga Pengelola Dana Pendidikan [LPDP]). DL is supported by the Wellcome Trust. For the purpose of Open Access, the author has applied a CC BY public copyright license to any Author Accepted Manuscript version arising from this submission. The funders of the investigators and study had no role in the study design, data collection, data analysis, data interpretation, or writing of the manuscript. The corresponding author had the final responsibility for the decision to submit for publication.

**Competing interests:** The authors have declared that no competing interests exist.

## Results

Of 1,311 COVID-19 and 58,917 non-COVID-19 inpatients, 333 (25.4%) and 18,837 (32.0%) received a parenteral antibiotic for at least four consecutive days. Proportion of patients having BC taken within ±1 calendar day of parenteral antibiotics being started was higher in COVID-19 than in non-COVID-19 patients (21.0% [70/333] vs. 18.7% [3,529/18,837]; p<0.001). Cumulative incidence of having a BC taken within 28 days was higher in COVID-19 than in non-COVID-19 patients (44.7% [149/333] vs. 33.2% [6,254/18,837]; adjusted subdistribution-hazard ratio [aSHR] 1.71, 95% confidence interval [CI] 1.47–1.99, p<0.001). The median time to discontinuation of parenteral antibiotics was longer in COVID-19 than in non-COVID-19 patients (13 days vs. 8 days; aSHR 0.73, 95%CI 0.65–0.83, p<0.001).

## Conclusions

Routine electronic data could be used to inform diagnostic and antibiotic use practices in LMICs. In Indonesia, the proportion of timely blood culture is low in both COVID-19 and non-COVID-19 patients, and duration of parenteral antibiotics is longer in COVID-19 patients. Improving diagnostic and antimicrobial stewardship is critically needed.

## Introduction

Antimicrobial-resistant (AMR) bacterial infection is a global threat to public health [1]. Low and middle-income countries (LMICs), including Indonesia, are considered hotspots of AMR, driven by the lack of laboratory support for infectious disease diagnosis and high levels of inappropriate antibiotic use in hospitals [2, 3].

Diagnostic and antimicrobial stewardship (AMS) are increasingly implemented in LMICs [4–6]. Although processes of the stewardship programmes (e.g., presence of standard operating procedures [SOPs] and how many patient medical charts are reviewed) are frequently monitored and reported, outcomes of those stewardship programmes (e.g. increase in specimens submitted for bacterial culture according to SOPs and defined daily dose [DDDs] per 1,000 patient-days) or patient outcomes (e.g. in-hospital mortality) are rarely monitored and reported. This is because measuring and analyzing diagnostic and antibiotic use practices overtime are labor-intensive and prohibitively costly.

In our recent report we found that the etiology of bloodstream infections (BSI) and proportion of AMR-BSI were similar between COVID-19 and non-COVID-19 patients hospitalized at the Indonesian national referral hospital, Jakarta, Indonesia, from 1 January 2019 to 31 December 2020, and that reported incidence rates of hospital-origin AMR-BSI increased in 2020, which was likely attributable to increased blood culture utilization [7]. Here, we aimed to analyze drug dispensing data, and evaluate diagnostic and antibiotic use practices in the Indonesian national referral hospital before and during the COVID-19 pandemic.

## Methods

### Study design, setting and population

We conducted a retrospective hospital-wide longitudinal surveillance study using routine electronic data on all patients hospitalized at Cipto Mangunkusumo National Hospital (CMNH), the national referral hospital in Jakarta, Indonesia, from 1 January 2019 to 31 December 2020.

## Data collection

We used the routine hospital databases containing hospital admission, laboratory microbiology and drug dispensing data. The hospital admission data collected included medical record number (MRN), sex, age, admission and discharge date. The microbiology laboratory data collected included MRN, admission date, specimen type, specimen collection date, culture result using conventional bacterial identification methods and Vitek®2 (bioMerieux, Inc. Durham, USA), antibiotic susceptibility profile according to Clinical and Laboratory Standards Institute guidelines, and result report date. The drug dispensing data collected included MRN, admission date, drug name, route of drug administration, the dosage regimen, drug start date and stop date. The MRN and admission date were used to link the three databases together. Data were accessed since 1 March 2021.

## Definitions

**Severe infection.** Patients with severe infection were defined using the criteria modified from the United States Centers for Disease Control and Prevention [8]. Patients who received a parenteral antibiotic for at least four consecutive days was used as a surrogate for severe infection, with the first calendar day equal to the start date of parenteral antibiotics. Patients who died, were discharged to a hospice or transferred to other hospital before completing four consecutive days of parenteral antibiotics and had parenteral antibiotics continuously until the day prior to death, hospice discharge or transfer were also included as patients with severe infection [9].

**Diagnostic practices.** To measure diagnostic practices, we estimated (a) the proportion of patients with blood culture (BC) taken within ±1 calendar day of parenteral antibiotics being started, (b) the proportion of patients with BC taken within 28 days of parenteral antibiotics being started, and (c) the median BC turnaround-time [4, 5, 9]. The proportion of patients having BC taken within ±1 calendar day of parenteral antibiotics being started was used to measure the compliance with the recommendations that blood specimens should be taken before starting antimicrobial therapy in patients with suspected sepsis and septic shock if no substantial delay in the start of antimicrobials occurs (i.e. <45 min) and that if the antibiotic is administered first, the blood specimens should be collected for culture within 24 hours [10]. The proportion of patients with BCs taken within 28 days was defined as the ratio of the number of patients with severe infection where BCs were taken from one calendar day before to 28 calendar days after starting parenteral antibiotics, and the total number of patients with severe infection. BC turn-around time was defined as the time interval between BC taken from patients and BC results positive for organisms reported by laboratory [4].

**Antibiotic use practices.** To measure antibiotic use practices, we estimated (a) the proportion of antibiotic use based on the AWaRe system (Access, Watch and Reserve) [5], (b) median time to discontinuation of parenteral antibiotics [5], (c) the proportion of patients with narrow-spectrum parenteral antibiotics (antibiotics within the Access and Watch category without anti-methicillin-resistant *Staphylococcus aureus* [MRSA] and antpseudomonal activity) within 2 calendar days following BC report among those who had a BC positive for 3rd-generation cephalosporin-sensitive *Escherichia coli* (3GCSEC) or *Klebsiella pneumoniae* (3GCSKP) and remained on a parenteral antibiotic on the day that the BC results were reported [11], and (d) the proportion of patients with parenteral antibiotic discontinuation within 2 calendar days following BC report among those who had BC results negative for organisms and remained on a parenteral antibiotic on the day that the BC results were reported [12].

Antibiotic use among inpatients was measured in terms of DDD per 1,000 inpatient-days. DDD was defined as the assumed average maintenance dose per day for a drug used for its main indication in adults as specified by World Health Organization (WHO) [13]. Pharmacological subgroups of antibacterials for systemic use was defined according to group based on the Anatomical Therapeutic Chemical classification system, including tetracyclines (J01A), amphenicols (J01B), beta-lactam antibacterials, penicillins (J01C), other beta-lactam antibacterials (J01D), sulfonamides and trimethoprim (J01E), macrolides, lincosamides and streptogramins (J01F), aminoglycoside antibacterials (J01G), quinolone antibacterials (J01M), combinations of antibacterials (J01R), other antibacterials (J01X), agents against amoebiasis and other protozoal diseases (P01A).

We also assessed the change of parenteral antibiotic among patients with severe infection within 28 days. Parenteral antibiotic use on each calendar day were categorized as escalation, de-escalation, change to other antibiotics with similar spectrum, no change from the previous antibiotic or stop. Antibiotic escalation was defined as (a) adding a new parenteral antibiotic, (b) changing parenteral antibiotic with an increase in AWaRe categories, or (c) changing parenteral antibiotic within the Watch category but from those without anti-MRSA and antipseudomonal activity to those with anti-MRSA activity (e.g. vancomycin) or antipseudomonal activity (e.g. antipseudomonal cephalosporin, antipseudomonal penicillin and carbapenems) compared to the previous day. De-escalation was defined as the inverse of these. Change to other antibiotics with similar spectrum was defined as changing parenteral antibiotics to other parenteral antibiotics within similar AWaRe categories, and with similar anti-MRSA activity and antipseudomonal activity for parenteral antibiotics within the Watch category compared to the previous day. No change was defined as continuation of current parenteral antibiotics, while stop was defined as discontinuation of parenteral antibiotics [12, 14–17]. We also evaluated initial parenteral antibiotics. Parenteral antibiotics being prescribed within the first calendar day that a parenteral antibiotic was started were regarded as initial parenteral antibiotics [18, 19].

## Ethics

The study was approved by the Faculty of Medicine Universitas Indonesia Ethics Committee (KET-115/UN2.F1/ETIK/PPM.00.02/2021) and Oxford Tropical Research Ethics Committee (503–22). The requirement for individual patient consent was waived. Additional permission was obtained from the Innovation and Intellectual Property Directorate CMNH, to use the routine hospital databases for research.

## Data analysis

Pearson's chi-squared test or Fisher's Exact test was used to compare categorical variables between groups when appropriate. Mann-Whitney test was used to compare continuous variables between groups. We compared time to mortality, having BC taken and discontinuation of parenteral antibiotics within 28 days between groups by using Fine and Grey models to account for competing risks [20, 21]. Mortality included in-hospital mortality and discharge to a hospice. Discharge to home was considered a competing risk for mortality. Mortality, discharge to home and discontinuation of parenteral antibiotics were considered a competing risk for having BC taken. For patients who had parenteral antibiotics up to the calendar date that they were discharged home, all parenteral antibiotics were considered discontinued on the discharge date. Mortality was considered a competing risk for discontinuation of parenteral antibiotics. All data analyses were performed using the STATA version 15.1 (StataCorp,

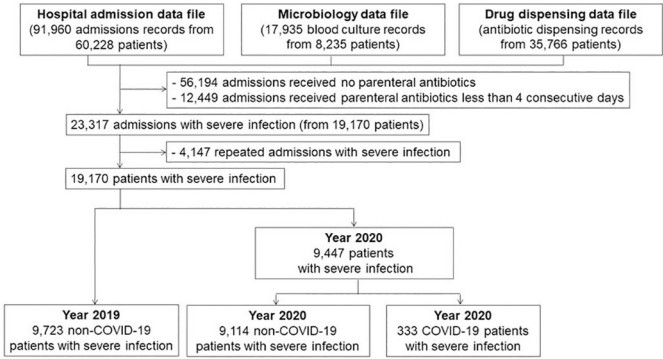

**Fig 1. Flow diagram.**

College Station, TX, USA). We visualized data using the GraphPad Prism version 8.3.0 (La Jolla, California, USA) and SankeyMATIC (https://sankeymatic.com/).

## Results

### Baseline characteristics

Of 91,960 admissions (from 60,228 patients) admitted from 1 Jan 2019 to 31 Dec 2020, 35,766 (38.9%) admissions received at least one antibiotic (Fig 1 and S1 Table). Total consumption of antibiotics among inpatients was 624.1 DDD per 1,000 inpatient-days, of which 77.6% (484.3 DDD per 1,000 inpatient-days) and 22.3% (139.8 DDD per 1,000 inpatient-days) were parenteral and oral antibiotics, respectively. Overall, 29.3% (182.4 DDD per 1,000 inpatient-days), 68.7% (429.3 DDD per 1,000 inpatient-days) and 2.0% (12.4 DDD per 1,000 inpatient-days) were classified as Access, Watch and Reserve categories, respectively (S1 Fig).

Of all parenteral antibiotic consumption among inpatients (484.3 DDD per 1,000 inpatient-days), 46.9% (227.1 DDD per 1,000 inpatient-days) were other beta-lactam antibacterials (J01D), 20.2% (97.8 DDD per 1,000 inpatient-days) were quinolone antibacterials (J01M), and 15.4% (74.6 DDD per 1,000 inpatient-days) were penicillins (J01C) (S2 Fig). Of all other beta-lactam antibacterials (J01D) consumption (227.1 DDD per 1,000 inpatient-days), 65.0% (147.6 DDD per 1,000 inpatient-days) were 3rd-generation cephalosporins (J01DD), 21.7% (49.3 DDD per 1,000 inpatient-days) were carbapenems (J01DH), and 8.0% (18.2 DDD per 1,000 inpatient-days) were 4th-generation cephalosporins (J01DE).

Of 91,960 admissions (60,228 patients), 1,373 (from 1,311 patients) were COVID-19 cases. Total antibiotic consumption was 1,502.8 and 606.3 DDD per 1,000 inpatient-days in COVID-19 and non-COVID-19 patients, respectively (S1 Table).

Of 60,228 patients, 19,170 (31.8%) received at least four consecutive days of parenteral antibiotics, were classified as patients with severe infection, and the first admission per patient which fulfilled the criteria of severe infection were included for further analysis.

### Patients with severe infection

Among 19,170 patients with severe infection, 333 (1.7%) were COVID-19 patients and 18,837 (98.3%) were non-COVID-19 patients (Table 1). The median age was higher in COVID-19 patients than non-COVID-19 patients (47 vs. 39 years, p<0.001). The sex distribution was not different between the two groups (p = 0.23).

**Table 1. Characteristics of 19,170 patients with severe infection.**

| Parameters | COVID-19 patients with severe infection (n = 333) | Non-COVID-19 patients with severe infection (n = 18,837) | P values |
|---|---|---|---|
| Median age (years old, IQR) | 47 (IQR 25–62, range 0–93) | 39 (IQR 15–56, range 0–95) | <0.001 |
| Sex | | | |
| Female | 156 (46.8%) | 9,445 (50.1%) | 0.23 |
| Male | 177 (53.2%) | 9,392 (49.9%) | |
| Median duration of hospital stay (days) | 8 (IQR 4–16, range 1–90) | 8 (IQR 4–15, range 1–119) | 0.08 |
| Time when a parenteral antibiotic was started [a] | | | |
| 1–2 calendar days after admission | 207 (62.2%) | 13,137 (69.7%) | <0.001 |
| 3–7 calendar days after admission | 97 (29.1%) | 3,840 (20.4%) | |
| 7 calendar days after admission | 29 (8.7%) | 1,860 (9.9%) | |
| Number of parenteral antibiotics received on the day that a parenteral antibiotic was started (%) | | | |
| 1 | 255 (76.6%) | 14,091 (74.8%) | 0.66 |
| 2 | 72 (21.6%) | 4,451 (23.6%) | |
| $\geq$3 | 6 (1.8%) | 295 (1.6%) | |
| Total antibiotic consumption (DDD per 1,000 patient-days) [b] | 1656.4 | 807.6 | |
| Access category | 153.9 (9.3%) | 263.8 (32.7%) | <0.001 |
| Watch category | 1427.3 (86.2%) | 520.3 (64.4%) | |
| Reserve category | 75.2 (4.5%) | 23.5 (2.9%) | |
| 28-day mortality [c] | 95 (28.5%) | 4,485 (23.8%) | 0.02 |

Only the first admission per patient which fulfilled the criteria of severe infection were included. Patients with severe infection were defined as those who received a parenteral antibiotic for at least four consecutive days. Patients who died, were discharged to a hospice or transferred to other hospital before completing four consecutive days of parenteral antibiotics and had parenteral antibiotics continuously until the day prior to death, hospice discharge or transfer were also included as patients with severe infection

[a] Time was measured when a parenteral antibiotic was started and continued for at least four consecutive days. Staring parenteral antibiotics on the day of admission was equal to 1 calendar day after admission.

[b] Total antibiotic consumption was estimated from the day of parenteral antibiotics being started to the discharge date.

[c] 28-day mortality included in-hospital mortality and discharged to a hospice within 28 days of parenteral antibiotics being started.

The proportion of parenteral antibiotics consumption under Watch and Reserve category was significantly higher in COVID-19 patients than non-COVID-19 patients (86.2% vs. 64.4% and 4.5% vs. 2.9%, respectively; p<0.001; Table 1). The proportion of patients being prescribed one, two or at least three initial parenteral antibiotics within the first calendar days of parenteral antibiotics being started was not different between two groups (p = 0.66) (Table 1). Ceftriaxone (36.9% [123/333]), levofloxacin (18.9% [63/333]) and meropenem (9.3% [31/333]) were commonly used as the initial antibiotics in COVID-19 patients, while ceftriaxone (30.9% [582/18,837]), ampicillin/sulbactam (8.5% [160/18,837]) and metronidazole (7.8% [147/18,837]) were commonly prescribed as the initial antibiotics in non-COVID-19 patients (S3 Fig).

The 28-day mortality among 19,170 patients with severe infection was 23.9% (Table 1 and S2 Table). Using Fine and Grey model, we found that the risk of mortality was higher in COVID-19 patients than non-COVID-19 patients (28.5% vs. 23.8%, adjusted subdistribution-hazard ratios [aSHR] 1.29; 95%CI 1.04–1.58, p = 0.02; Table 2 and S4A Fig).

## Blood culture practices

Of 19,170 patients with severe infection, 3,599 (18.8%) had BC taken within ±1 calendar day of parenteral antibiotics being started (Table 3 and S5 Fig). The proportion was higher in

**Table 2. Multivariable analyses for time to mortality, having blood culture sampled and discontinuation of parenteral antibiotics among 19,170 patients with severe infection.**

| Variables | Mortality [a] | | Having blood culture sampled [b] | | Discontinuation of parenteral antibiotics [c] | |
|---|---|---|---|---|---|---|
| | Adjusted SHR | P values | Adjusted SHR | P values | Adjusted SHR | P values |
| **COVID-19 status** | | | | | | |
| Non-COVID-19 | Reference | | Reference | | Reference | |
| COVID-19 | 1.29 (1.04–1.58) | 0.02 | 1.71 (1.47–1.99) | <0.001 | 0.73 (0.65–0.83) | <0.001 |
| **Sex** | | | | | | |
| Female | Reference | | Reference | | Reference | |
| Male | 1.02 (0.97–1.09) | 0.38 | 1.04 (0.99–1.09) | <0.001 | 0.92 (0.89–0.94) | <0.001 |
| **Age (years old)** | | | | | | |
| <1 | 1.09 (0.93–1.28) | <0.001 | 7.54 (6.73–8.45) | <0.001 | 0.84 (0.79–0.89) | 0.72 |
| 1–4 | 1.01 (0.84–1.20) | | 4.54 (4.01–5.14) | | 0.88 (0.82–0.95) | |
| 5–14 | 1.07 (0.92–1.25) | | 3.98 (3.52–4.51) | | 0.94 (0.87–1.00) | |
| 15–24 | 1.26 (1.07–1.48) | | 1.90 (1.65–2.17) | | 0.88 (0.82–0.94) | |
| 25–34 | Reference | | Reference | | Reference | |
| 35–44 | 1.24 (1.07–1.41) | | 1.25 (1.09–1.43) | | 0.99 (0.93–1.05) | |
| 45–54 | 1.23 (1.07–1.41) | | 1.51 (1.33–1.71) | | 0.94 (0.89–1.00) | |
| 55–64 | 1.22 (1.06–1.40) | | 1.51 (1.33–1.72) | | 0.90 (0.85–0.95) | |
| ≥65 | 1.30 (1.13–1.49) | | 1.44 (1.26–1.64) | | 0.87 (0.82–0.92) | |
| **Admission year** | | | | | | |
| 2019 | Reference | | Reference | | Reference | |
| 2020 | 0.96 (0.91–1.02) | 0.18 | 1.06 (1.01–1.11) | 0.02 | 0.94 (0.92–0.97) | <0.001 |

SHR = sub-distribution hazard ratio. Fine and Grey sub-distribution hazard models were used.

[a] Mortality included in-hospital mortality and discharged to a hospice. Discharged home from the hospital was considered as a competing risk for mortality.

[b] In-hospital mortality, discharged to a hospice and discontinuation of parenteral antibiotics were considered as a competing risk for having blood culture sampled. Blood culture sampled on -1 calendar day when a parenteral antibiotic was started was included in the model as sampled on the calendar day when a parenteral antibiotic was started.

[c] In-hospital mortality and discharged to a hospice were considered as a competing risk for discontinuation of parenteral antibiotics.

COVID-19 than in non-COVID-19 patients (21.0% [70/333] vs. 18.7% [3,529/18,837]; p<0.001).

The proportion of patients having BC taken within 28 days after parenteral antibiotics being started was 33.4%. Using Fine and Grey model, we found that having BC taken within 28 days was higher in COVID-19 than non-COVID-19 patients (44.7% vs. 33.2%; aSHR 1.71; 95%CI 1.47–1.99, p<0.001; Table 2) and in those admitted in 2020 than those admitted in 2019 (aSHR 1.06; 95%CI 1.01–1.11, p = 0.02). S4B Fig shows that, of COVID-19 and non-COVID-19 patients, 21.0% and 18.7% had the first BC taken on ±1 calendar day of parenteral antibiotics being started and the other 23.7% and 14.5% gradually had the first BC taken between calendar day 2 and 14 of parenteral antibiotics being started, respectively.

The median BC turn-around time was not different between the two groups (4 vs. 4 days; p = 0.69).

## Antibiotic use practices

Fig 2 shows escalation, change, de-escalation and discontinuation of parenteral antibiotics compared to the parenteral antibiotics the patients received the day before. Of 19,170 patients with severe infection, 2,466 (12.9%) had parenteral antibiotics escalated, 3,267 (17.0%)

**Table 3. Parameters for clinical diagnostic practices and antibiotic use practices among 19,170 patients with severe infection.**

| Parameters | COVID-19 patients with severe infection (n = 333) | Non-COVID-19 patients with severe infection (n = 18,837) | P values |
|---|---|---|---|
| **Parameters for clinical diagnostic practices** | | | |
| Proportion of patients with BC taken within ±1 calendar day of parenteral antibiotic being started | 21.0% (70/333) | 18.7% (3,529/18,837) | <0.001 |
| Proportion of patients with BC taken from calendar day 3 to day 28 of parenteral antibiotic being started | 31.8% (106/333) | 21.3% (4,012/18,837) | <0.001 |
| Proportion of patients with at least one BC taken within 28 calendar days of parenteral antibiotic being started | 44.7% (149/333) | 33.2% (6,254/18,837) | <0.001 |
| Median BC turn-around time (days, IQR, n) | 4 (4–5, n = 34) | 4 (4–5, n = 1,593) | 0.69 |
| **Parameters for antibiotic use practices** | | | |
| Proportion of patients with narrow-spectrum parenteral antibiotic within 2 calendar days following BC report among those who had a BC positive for 3GCSEC or 3GCSKP and remained on a parenteral antibiotic on the day that BC results were reported | 0% (0/1) | 38.4% (20/52) | 0.43 |
| Proportion of patient with parenteral antibiotic discontinuation within 2 calendar following BC report among those who had a negative BC and remained on a parenteral antibiotic on the day that BC results were reported | 19.4% (7/36) | 26.8% (616/2,298) | 0.32 |

BC = blood culture, 3GCSEC = 3<sup>rd</sup> generation cephalosporin sensitive *E. coli*, 3GCSKP = 3<sup>rd</sup> generation cephalosporin sensitive *K. pneumoniae*, Narrow-spectrum parenteral antibiotics were defined as antibiotics within the Access and Watch category without anti-MRSA and antipseudomonal activity

changed to other antibiotics with similar spectrum, and 1,001 (5.2%) de-escalated at least once within 28 days after parenteral antibiotics being started. S6 Fig shows the proportion of the dispensed parenteral antibiotics categorized by AWaRe classification over 28 calendar days.

Of 19,170 patients with severe infection, 72.7% had parenteral antibiotics discontinued within 28 days (S4C Fig). The median time to discontinuation of parenteral antibiotics was longer in COVID-19 than non-COVID-19 patients (13 days vs. 8 days; aSHR 0.64; 95%CI

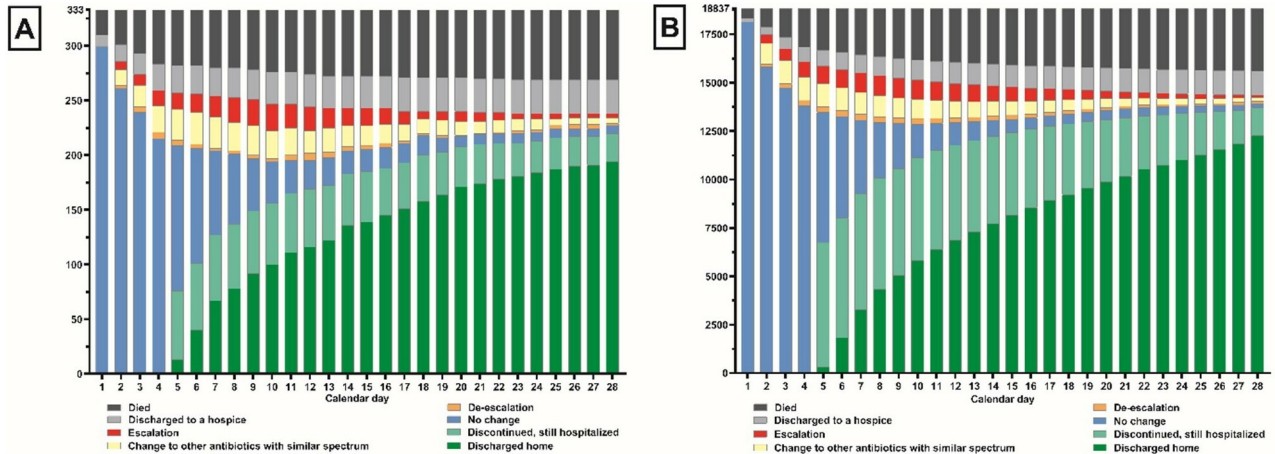

**Fig 2.** Antibiotic stewardship among (A) 333 COVID-19 patients and (B) 18,837 non-COVID-19 patients with severe infection. "Antibiotic escalation" was defined as (a) adding a new antibiotic, (b) changing antibiotics to other antibiotics with an increase in Access, Watch and Reserve categories, or (c) changing antibiotics in the Watch category without anti-MRSA and antipseudomonal activity to other antibiotics within the Watch category with anti-MRSA activity (e.g. vancomycin) or antipseudomonal activity (e.g. antipseudomonal cephalosporin, antipseudomonal penicillin and carbapenems) compared to the parenteral antibiotics received on the day before. "Antibiotic de-escalation" was defined as the inverse of these. "Change to other antibiotics with similar spectrum" was defined as changing antibiotics to other antibiotics within similar Access, Watch and Reserve categories, and with similar anti-MRSA activity and antipseudomonal activity for antibiotics within the Watch category. Only parenteral antibiotics were included in the analysis.

0.55–0.75, p<0.001; Table 2 and S4C Fig), and in those admitted in 2020 than those admitted in 2019 (aSHR 0.94; 95%CI 0.92–0.97, p<0.001).

Of 53 patients with severe infection who had a BC positive for 3GCSEC or 3GCSKP and remained on a parenteral antibiotic on the day that BC results were reported, 20 (37.7%) had narrow-spectrum parenteral antibiotic within 2 calendar days following BC report. The proportion was not different between the two groups (p = 0.43). Of 2,344 patients with severe infection who had BC results negative for any organisms and remained on a parenteral antibiotic on the day that BC results were reported, 623 (26.6%) had parenteral antibiotics discontinued within 2 calendar days following BC report. This proportion was not different between the two groups (p = 0.32).

## Discussion

This study demonstrates that the routine electronic hospital databases could be used to monitor and report outcomes of diagnostic and antimicrobial stewardship programmes in hospitals in LMICs. In the study hospital, we found that the proportion of timely BC was low in inpatients, and duration of parenteral antibiotics was longer in COVID-19 patients. Most of the parenteral antibiotics used for inpatients were from the Watch Category. After adjusting for COVID-19 patients, we found that the BC practice minimally improved in 2020 compared to 2019; however, the time to discontinuation of parenteral antibiotics was longer in 2020 compared to 2019. We shared these findings with the AMS committee of the hospital, highlighting the urgency of improving both diagnostic and antimicrobial stewardship programmes in the study hospital.

The proportion of patient with severe infection having BC sampled within ±1 calendar day of parenteral antibiotics being started was lower than those reported in other high-income countries [22, 23] and in Thailand [9]. Furthermore, we observed a high proportion of patients with severe infection having the first BC taken from calendar day 2 to 14 after the start of parenteral antibiotic administration. This shows that delayed BC is not uncommon in the study hospital, although there is an increasing trend of blood culture utilization during the study period [7]. This is of concern as underutilization and delayed BC can have a negative impact on individual patient management [4] and can overestimate proportions and underestimate incidence rates of AMR infections in the cumulative antibiograms [9, 24].

The better BC practice among COVID-19 patients compared to non-COVID-19 patients is encouraging. This could be because our hospital, which is the national referral hospital, manages COVID-19 patients with comorbidities and severe clinical presentations. Additionally, in COVID-19 patients, clinicians tend to consult infectious diseases physicians and sampled BC prior to parenteral antibiotic administration more frequently [7]. This improvement should be strengthened further until all patients have BC sampled within ±1 calendar day of parenteral antimicrobial therapy being started [25].

We observed a longer median BC turn-around time compared to those reported in high-income countries, i.e. 2–3 days [26–28]. The BC turn-around time is critical for gaining the maximum benefits of BC as a guidance for clinicians in deciding rational antibiotic prescriptions, reduce length of hospitalization and increase patient survival [4, 29–33]. Efforts to shorten BC turn-around time by improvement of internal laboratory workflows (e.g., work shifts and laboratory staffing) should be commenced [4, 28, 32, 34].

The longer median time to discontinuation of parenteral antibiotics was observed in inpatients compared to those recommended in the guidelines (i.e. 5–8 days) [10]. The longer time to discontinuation parenteral antibiotics and higher use of Watch and Reserve antibiotics in COVID-19 patients could be due to the fear of high mortality in the early COVID-19 pandemic, in addition to the clinical profile of patients managed in our hospital [35]. Previous

studies reported an overuse of antibiotics amongst COVID-19 hospitalized patients. There might be a false belief of antibiotic use as medical prophylaxis to reduce secondary bacterial infections [36–39]. These highlight the need to improve clinician adherence to the well-developed evidence-based antibiotic guidelines [38, 39].

We observed that a de-escalation practice is uncommon in our study hospital using multiple parameters, similar to reports from other LMICs [40]. The common barrier to de-escalation following BC positive for monomicrobial narrow-spectrum antibiotic susceptible Enterobacterales could be the fear that patients may deteriorate if the broad-spectrum antibiotic is de-escalated [11]. The low proportion of antibiotic discontinuation following negative BC could reflect the physician lack of trust in negative results [14]. These data highlight the necessity to emphasize consideration of de-escalation following positive BC with drug sensitive pathogens and discontinuation following negative BC in patients with positive treatment response as an opportunity to reduce the overuse of broad-spectrum antibiotics [12, 14, 41].

Our study has several limitations. First, we could not exclude patients who had parenteral antibiotics for surgical prophylaxis for at least four consecutive days and analyze data for each clinical syndrome due to data limitation. Second, parameters used for evaluating practices are not free from bias, although they have been frequently used in previous studies. Third, use of routine data could not determine whether the practice in every single patient was appropriate or inappropriate. However, the analysis of routine data can represent the overall practice at the study hospital. Future studies could compare the routine data among hospitals of similar bed count and over time, or use the routine data to select specific sets of patients for further retrospective chart reviews. Lastly, the findings may not be generalizable to all hospitals in LMICs.

## Conclusions

In the Indonesian referral hospital, the proportion of timely BC is low, and duration of parenteral antibiotics is long in both COVID-19 and non-COVID-19 patients. Improving diagnostic and antimicrobial stewardship is critically needed. We recommend hospitals in LMICs to perform routine monitoring and improvement of diagnostic and antimicrobial stewardship.

## Supporting information

**S1 Fig. Proportional consumption of (A) parenteral and (B) oral antibiotics by AWaRe categorization among all admissions (n = 91,960) of all inpatients (n = 60,228) between 2019 and 2020.** For this figure, antibiotics in the Watch category was divided to Watch and Watch+. Watch+ category comprises antibiotics in the Watch category with anti-MRSA activity (e.g. vancomycin) or antipseudomonal activity (e.g. antipseudomonal cephalosporin, antipseudomonal penicillin and carbapenems).
(DOCX)

**S2 Fig. Consumption of parenteral antibiotics (DDD per 1,000 patient days) by pharmacological subgroups among all admissions (n = 91,960) of all inpatients (n = 60,228) between 2019 and 2020.** J01A: tetracyclines; J01B: amphenicols; J01C: beta-lactam antibacterials, penicillins; J01D: other beta-lactam antibacterials; J01E: sulfonamides and trimethoprim; J01F: macrolides, lincosamides and streptogramins; J01G: aminoglycoside antibacterials; J01M: quinolone antibacterials; J01R: combinations of antibacterials; J01X: other antibacterials, P01A: agents against amoebiasis and other protozoal diseases.
(DOCX)

**S3 Fig. Distribution of initial antibiotic among 19,170 patients with severe infection.** Parenteral antibiotics being prescribed within the first calendar day that a parenteral antibiotic

was started were regarded as initial parenteral antibiotics. Patients who received a parenteral antibiotic for at least four consecutive days was used as a surrogate for severe infection, with the first calendar equal to the start date of parenteral antibiotics. Patients who died, were discharged to a hospice or transferred to other hospital before completing four consecutive days of parenteral antibiotics and had parenteral antibiotics continuously until the day prior to death, hospice discharge or transfer were also included as patients with severe infection. (DOCX)

**S4 Fig.** Cumulative incidence of (A) mortality, (B) having blood culture sampled, and (C) discontinuation of parenteral antibiotics among 19,170 patients with severe infection. (DOCX)

**S5 Fig. Sankey diagram showing first blood culture test and blood culture result among 19,170 patients with severe infection.** (DOCX)

**S6 Fig. Proportional consumption of parenteral antibiotics given to (A) 333 COVID-19 patients and (B) 18,837 non-COVID-19 patients with severe infection by AWaRe categorization over 28 calendar days.** For this figure, antibiotics in the Watch category was divided to Watch and Watch+. Watch+ category comprises antibiotics in Watch category with anti-MRSA activity (e.g. vancomycin) or antipseudomonal activity (e.g. antipseudomonal cephalosporin, antipseudomonal penicillin and carbapenems. (DOCX)

**S1 Table. Baseline characteristics of all patients hospitalized at the Indonesian national referral hospital, Jakarta, Indonesia from 1 January 2019 to 31 December 2020.** (DOCX)

**S2 Table. Mortality, proportion of having blood culture sampled and median time to parenteral antibiotics discontinuation of patients with severe infection within 28 days after parenteral antibiotics being started, stratified by variables and COVID-19 status.** (DOCX)

## Acknowledgments

We thank the Hospital Information System Management Unit, Cipto Mangunkusumo National Referral Hospital for the support for this project. We thank Samuel Susanto for technical assistance.

## Author Contributions

**Conceptualization:** Robert Sinto, Khie Chen Lie, Siti Setiati, Direk Limmathurotsakul.

**Data curation:** Robert Sinto, Khie Chen Lie, Siti Setiati, Direk Limmathurotsakul.

**Formal analysis:** Robert Sinto, Direk Limmathurotsakul.

**Funding acquisition:** Robert Sinto.

**Investigation:** Robert Sinto, Khie Chen Lie, Siti Setiati, Direk Limmathurotsakul.

**Methodology:** Robert Sinto, Khie Chen Lie, Siti Setiati, Direk Limmathurotsakul.

**Project administration:** Robert Sinto.

**Resources:** Robert Sinto.

**Software:** Robert Sinto.

**Supervision:** Robert Sinto, Khie Chen Lie, Siti Setiati, Direk Limmathurotsakul.

**Validation:** Robert Sinto, Sumariyono Sumariyono, Direk Limmathurotsakul.

**Visualization:** Robert Sinto, Direk Limmathurotsakul.

**Writing – original draft:** Robert Sinto, Direk Limmathurotsakul.

**Writing – review & editing:** Robert Sinto, Khie Chen Lie, Siti Setiati, Suhendro Suwarto, Erni J. Nelwan, Mulya Rahma Karyanti, Anis Karuniawati, Dean Handimulya Djumaryo, Ari Prayitno, Sumariyono Sumariyono, Mike Sharland, Catrin E. Moore, Raph L. Hamers, Nicholas P. J. Day, Direk Limmathurotsakul.

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
