## [Decision Letter · Decision Letter 0]

4 Jan 2024

Diagnostic and antibiotic use practices among COVID-19 and non-COVID-19 patients in the Indonesian National Referral Hospital

PONE-D-23-35454

Dear Dr. Sinto,

We’re pleased to inform you that your manuscript has been judged scientifically suitable for publication and will be formally accepted for publication once it meets all outstanding technical requirements.

Kind regards,

Arianit Jakupi, PhD

Academic Editor

PLOS ONE

Reviewers' comments:

Reviewer's Responses to Questions

**Comments to the Author**

1. Is the manuscript technically sound, and do the data support the conclusions?

Reviewer #1: Yes

Reviewer #2: Yes

2. Has the statistical analysis been performed appropriately and rigorously? 

Reviewer #1: Yes

Reviewer #2: Yes

3. Have the authors made all data underlying the findings in their manuscript fully available?

Reviewer #1: Yes

Reviewer #2: Yes

4. Is the manuscript presented in an intelligible fashion and written in standard English?

Reviewer #1: Yes

Reviewer #2: Yes

5. Review Comments to the Author

Reviewer #1: The manuscript delineates a scientifically robust investigation characterized by technical precision, underpinned by a judicious presentation of data substantiating the derived conclusions. The experiments conducted exhibit a rigorous methodology, incorporating meticulous attention to detail through the implementation of suitable controls and consideration of optimal sample sizes. The inferences drawn within the manuscript are judiciously founded upon the discerning analysis of the data proffered, thereby enhancing the overall scientific rigor of the study.

Reviewer #2: The manuscript demonstrates a high degree of technical soundness. The research methodology appears well planned and meticulously executed. The experimental design, data collection, and analysis are all appropriately documented. The study's objectives are clear, and the research questions are logically addressed. The use of appropriate methods and techniques in data collection and analysis enhances the manuscript's credibility.

6. PLOS authors have the option to publish the peer review history of their article (what does this mean?). If published, this will include your full peer review and any attached files.

Reviewer #1: No

Reviewer #2: No

---

## [Editor Report · Acceptance letter]

27 Feb 2024

PONE-D-23-35454 

PLOS ONE

Dear Dr. Sinto, 

I'm pleased to inform you that your manuscript has been deemed suitable for publication in PLOS ONE. Congratulations! Your manuscript is now being handed over to our production team.

Kind regards, 

on behalf of

Dr Arianit Jakupi 

Academic Editor

PLOS ONE